A quantitative assessment of site-level factors in influencing Chukar (Alectoris chukar) introduction outcomes

http://orcid.org/0000-0003-2302-6417 Smith Austin M. 1 2 amsmith11@usf.edu
http://orcid.org/0000-0001-7851-7382 Cropper, Jr. Wendell P. 3
http://orcid.org/0000-0003-4041-9343 Moulton Michael P. 4
1 Department of Integrative Biology, University of South Florida , Tampa, Florida , United States
2 School of Natural Resources and Environment, University of Florida , Gainesville, Florida , United States
3 School of Forest Resources and Conservation, University of Florida , Gainesville, Florida , United States
4 Department of Wildlife Ecology and Conservation, University of Florida , Gainesville, Florida , United States
Pimm Stuart
Electronic publication date: 2021 Apr 16
Publication date: 2021
Volume: 9
Electronic Location ID: e11280
Received 2020 Nov 11; Accepted 2021 Mar 24
Copyright: © 2021 Smith et al.
Copyright year: 2021
Copyright holder: Smith et al.
License: This is an open access article distributed under the terms of the Creative Commons Attribution License, which permits unrestricted use, distribution, reproduction and adaptation in any medium and for any purpose provided that it is properly attributed. For attribution, the original author(s), title, publication source (PeerJ) and either DOI or URL of the article must be cited.
License URL: https://creativecommons.org/licenses/by/4.0/

Keywords: Alectoris chukar, Bird introductions, Species distribution model, Foreign Game Investigation Program

Funding: The authors received no funding for this work.

==============================
Chukar partridges (Alectoris chukar) are popular game birds that have been introduced throughout the world. Propagules of varying magnitudes have been used to try and establish populations into novel locations, though the relationship between propagule size and species establishment remains speculative. Previous qualitative studies argue that site-level factors are of importance when determining where to release Chukar. We utilized machine learning ensembles to evaluate bioclimatic and topographic data from native and naturalized regions to produce predictive species distribution models (SDMs) and evaluate the relationship between establishment and site-level factors for the conterminous United States. Predictions were then compared to a distribution map based on recorded occurrences to determine model prediction performance. SDM predictions scored an average of 88% accuracy and suitability favored states where Chukars were successfully introduced and are present. Our study shows that the use of quantitative models in evaluating environmental variables and that site-level factors are strong indicators of habitat suitability and species establishment.

Introduction

A central goal in the study of introduced species involves determining the forces that might influence introduction success. Duncan, Blackburn & Sol (2003) argued that such forces fall in to three categories: species-level factors; site-level factors; and event-level factors. In a recent analysis, Moulton et al. (2018), examined a large database of game bird introductions to the United States (US), from the Foreign Game Investigation Program (FGIP). Based on the pattern of introduction successes, Moulton et al. (2018), argued that the best explanation for the pattern indicated that location-level factors must frequently be more important in deciding the fates of game bird introductions than the often-championed event-level factor of propagule pressure (e.g., Williamson & Fitter, 1996; Lockwood, Cassey & Blackburn, 2005). However, the evidence for an important role for location-level factors was indirect, and non-specific.

In the FGIP database, 13 of the 17 species released invariably failed to establish populations in the US despite a wide range of propagule sizes. The remaining four species included the Chukar (Alectoris chukar); Common Pheasant (Phasianus colchicus); Gray Partridge (Perdix perdix); and Himalayan Snowcock (Tetraogallus himalayensis). Of these species, we selected the Chukar for a detailed study of location-level factors in predicting the fate of introductions.

Chukars are gallinaceous birds that are found at higher altitudes in arid regions consisting of talus slopes, short grasses, and shrubs (Alcorn & Richardson, 1951; Barnett, 1952; Galbreath & Moreland, 1953; Christensen, 1954, 1970, 1996; Bohl, 1957; Harper, Harry & Bailey, 1958; Tomlinson, 1960). Their native range includes southeastern Europe, central and west Asia, and the Himalayas, but have been successfully introduced to North America, the Hawaiian Islands, and New Zealand (Christensen, 1970, 1996; Long, 1981; Lever, 1987). In the conterminous US, Chukars are well-established in 10 western states; their distribution is centered in the Great Basin and extends to eastern Washington, northern Idaho, western Wyoming and Colorado, the northwestern border of Arizona, and parts of Montana (Christensen, 1970, 1996). Efforts have been made to establish Chukars in several other parts of world, including Australia, western Europe, and southern Africa; however, most attempts were unsuccessful (Long, 1981; Lever, 1987).

A number of sources remain inconsistent, specifically with what constitutes as an introduction or release attempt. In some cases, an introduction is identified as a single release or instance, while others are a summation of releases over some duration of time (Moulton & Cropper, 2020). Even then, large propagules may have occurred not out of necessity, but rather to expedite population densities (Moulton, Cropper & Broz, 2015; Moulton et al., 2018; Moulton & Cropper, 2016). Spatial scale is also often ignored, and previous studies reviewed introductions at a state or country level rather than more specific locations (Moulton & Cropper, 2020). Therefore, three unverifiable assumptions are made from this: individuals were only released in areas considered to be suitable; propagule size was determined by the amount of available habitat; and introductions were distributed uniformly, and not in clustered locations.

Gullion (1965) criticized these assumptions when reviewing the FGIP data and questioned if birds truly adapted/acclimated to new geographical territory, or if they were introduced and succeeded in locations that were most similar to native habitat. He further asserted that if an introduced species were to establish in a new location, that should hold true for even small propagules.

The most well-documented experimental programs occurred in the US, with both state and federal game commissions releasing individuals in a variety of habitats (e.g., Nagel, 1945; Galbreath & Moreland, 1953; Long, 1981; Lever, 1987). Bump (1968) reported that probably every state within the US attempted to establish Chukars; fortunately, a great deal of detailed environmental information is available as well as general locations within each state where releases were successful and unsuccessful (Nagel, 1945; Alcorn & Richardson, 1951; Barnett, 1952; Galbreath & Moreland, 1953; Christensen, 1954, 1970, 1996; Bohl, 1957; Harper, Harry & Bailey, 1958; Long, 1981; Lever, 1987). These qualitative assessments indicate that Chukars favored habitats that resembled their native range, but they might not persist in environments with too much snow (e.g., Gullion, 1965; Christensen, 1970), or hot, arid regions with limited access to free water (e.g., Galbreath & Moreland, 1953; Tomlinson, 1960; Christensen, 1970; Larsen et al., 2007).

Although Chukars have been studied extensively (e.g., Nagel, 1945; Alcorn & Richardson, 1951; Barnett, 1952; Galbreath & Moreland, 1953; Christensen, 1954, 1970, 1996; Bohl, 1957; Harper, Harry & Bailey, 1958; Tomlinson, 1960; Moulton, Cropper & Broz, 2015; Moulton et al., 2018; Moulton & Cropper, 2016, 2019, 2020), little information exists for quantitative comparisons of locations deemed suitable versus not suitable. With this in mind, the goal of our study was to develop quantitative models using machine learning algorithms to assess environmental factors of sites in the Chukar’s native range, and to determine/predict most suitable sites for introductions in the 48 contiguous states of the US.

Such algorithms represent a standard practice for formulating a species distribution model or SDM, which identifies characteristics of a species niche or could be applied to predicting potential range expansions or contractions (Phillips et al., 2009; Elith et al., 2011; Hijmans, 2012; Hijmans & Elith, 2017). Species distribution models (SDMs) use species occurrence records and compare them to areas where they are absent. In many cases, absences were not recorded, and models refer to ‘pseudo-absent’ or ‘background’ data, places assumed less suitable due to the lack of occurrence records, for comparative purposes (Phillips et al., 2009). These models score and rank each point in the area of interest to determine the level of suitability. Traditional methods rely on regression techniques such as generalized linear models and generalized additive models because of their seemingly straightforward interpretability (Elith, Leathwick & Hastie, 2008; Hastie, Tibshirani & Friedman, 2009). Machine learning algorithms are also able to produce such models but do not require any preconceived relationships between environmental covariates and are able to handle complex data distributions (Elith, Leathwick & Hastie, 2008; Hastie, Tibshirani & Friedman, 2009; Elith et al., 2011). A common problem when trying to quantify habitat quality is determining which variables to include in model building given that reducing the dimensionality of the problem could lead to missing subtle non-linear interactions. We chose algorithms with proven success when applied to large, multi-dimensional data sets to remove bias from the covariate selection. A common practice is to choose the ‘best’ model based on a set of statistics. To eliminate this bias and to ensure a collective analysis, we chose to construct and examine ensembles (e.g., model averages) to assess the importance of various site-level factors.

Methods

All of our analysis, model construction, and graphics were done using R Ver. 3.5.2 statistical computing language (R Core Team, 2018). All data points and covariates were measured using geospatial packages and extracted from 2.5-min spatial scale raster and polygon layers.

Species occurrence data

We constructed two sets of models for our analysis. First, we collected observations submitted to eBird (Sullivan et al., 2009; eBird, 2021) that provided a media source for validation. We removed all duplicate occurrences that shared longitudinal-latitudinal coordinates leaving us with 1,302 occurrences. We compared these occurrences to where Chukars were successfully introduced and determined they were appropriate for our analysis since both records shared similar distributions (e.g., Christensen, 1970, 1996; Long, 1981; Lever, 1987). Note, 21 occurrences that met our requirements were from states where Chukar failed to establish (Christensen, 1970, 1996; Long, 1981; Lever, 1987). While there are no records that indicate these are wild Chukar or that the locations are suitable, we retained these points to avoid sampling bias. We then randomly selected 10,000 background points from all terrestrial regions, excluding Antarctica and snow-covered Greenland using the ‘spsample’ function from the ‘sp’ package (Pebesma & Bivand, 2005). We chose to exclude these regions because the persistent, thick snow-covered areas are unsuitable habitat for Chukar, as noted above. Bohl (1957) documented several studies within the US regarding the distance travelled by released Chukars. We averaged these measurements (μ=49km) and created circular buffers around each point and merged these polygons to create an estimated range plot (Fig. 1).

Figure 1 Chukar eBird occurrences and estimated range model.

(A) World distribution of eBird Chukar occurrences that provided a media source. (B) Estimated naturalized range for the contiguous United States with the sample of California occurrences used to build models.

Mori et al. (2019) note that subspecies selection may account for variability in distribution models. Therefore, we chose to create similar models using the 97 points from the naturalized region in California, US (Fig. 1) and predict the remaining area of the conterminous US. We chose this region because historical records show that California was one of the first places where Chukars were imported from their native range, reared, released, established/acclimated, and hunted in the US. Furthermore, several other state game commissions acquired Chukars from California game farms, with both failed and successful introductions of large propagules documented (Nagel, 1945; Galbreath & Moreland, 1953, Harper, Harry & Bailey, 1958). It should be noted that though Chukars in Nevada, the first state in the US with successfully established Chukars and the first to hold a hunting season, were not derived from California game stock, but were the same subspecies as those in California (Alcorn & Richardson, 1951; Harper, Harry & Bailey, 1958; Christensen, 1970).

Model covariates

Choosing appropriate model covariates often depends on spatial scale (Pearson & Dawson, 2003; Luoto, Virkkala & Heikkinen, 2006). Some studies (Pearson & Dawson, 2003; Thuiller, Araújo & Lavorel, 2004; Luoto, Virkkala & Heikkinen, 2006; Engler et al., 2017) note that while finer spatial scales may account for more specific habitat characteristics (e.g., biotic interactions, species dispersal limitations, soil and land cover types), Pearson & Dawson (2003) suggest a hierarchal consideration of factors, with importance dependent of spatial scale. Since our study was of the subcontinental scale and a coarser spatial resolution, model parameters pertained to climate and physiography. A favored set of measurements for SDMs are the 19 WorldClim bioclimatic variables (Hijmans et al., 2005; Fick & Hijmans, 2017; Schatz, Kramer & Drake, 2017) and were included in our model building procedure. Limiting studies to these variables alone has been criticized as this might overestimate habitat ranges and does not speak to local ecosystems (Hirzel & Le Lay, 2008). Because Chukars require steep, talus slopes and favor higher altitudes (e.g., Christensen, 1970), we also included elevation, and calculated the slope, aspect, and Terrain Roughness Index score for each cell using the NASA Shuttle Radar Topography Mission raster layer provided by WorldClim (Fick & Hijmans, 2017). Finally, to ensure equal weight to all model covariates, we normalized each raster value prior to point extraction, where values range from 0 to 1.

Algorithms

An extensive study by Norberg et al. (2019) showed SDM predictions may vary due to a variety of factors including preconceived assumptions, statistical inference, and algorithmic framework, and encourage the use of several models rather than a single ‘best’ model. We, therefore, chose to incorporate several of their suggestions into our modeling framework.

We built our models using the ‘dismo’ package in R (Hijmans & Elith, 2017; Hijmans et al., 2017). We used the machine learning algorithms cited in Hijmans & Elith (2017) as these algorithms are widely used but have been shown to produce highly accurate models. These algorithms include artificial neural networks (ANN) as defined by Kulhanek, Leung & Ricciardi (2011), gradient boosting trees (GBM; Elith, Leathwick & Hastie, 2008), maximum entropy (MaxEnt; Elith et al., 2011), random forest (RF; Breiman, 2001), and support vector machines (SVM; Drake, Randin & Guisan, 2006).

A common statistical technique for model training and testing is to use a K-fold cross validation on the data to determine model performance (Hastie, Tibshirani & Friedman, 2009; Schatz, Kramer & Drake, 2017; Norberg et al., 2019). We used a 5-fold cross validation for all sets of models where all background points were used for each fold to enhance habitat variability.

To evaluate each iteration of model building, we chose to use the Area under the Receiver operating Curve (AUC) as a measure of model usefulness and testing performance for each fold. To test overall performance of each algorithm, we calculated the mean and standard deviation across all five folds, then created an ensemble of the model by averaging the prediction models. To evaluate model predictions, we transformed our ranked models to a binary map (i.e., suitable-not suitable) using the optimized specificity-sensitivity threshold measurement (Liu, Newell & White, 2016). We calculated the percent classification accuracy, sensitivity, and specificity of each model via confusion matrix comparing the predictive plots to the estimated naturalized range plot (Fig. 1) to determine model performance.

Finally, we created three ensembles for each algorithm, one average and two election models, and two collective election ensembles built using all 25 models. Here, an election is one where each model casts a ‘vote’ on the status of the raster cell and the classification in the final model is determined by a set proportion. For each algorithm, we calculated the average predictive value for each raster cell across the five folds and average the thresholds produce our binary classifications. For our election models we used a majority vote (MV, 3 of 5 folds) and a unanimous decision (UD), cases where all 5 folds agree, to determine suitability. We also used the majority (13 of 25 folds) and the UD frameworks for the two collective election ensembles. We then calculated our prediction statistics for comparative purposes.

Results

Modeling native points

We compared the training AUC scores of the model folds to determine model quality (Fig. 2). For the cross-validation testing scores, RF (μ=0.9884;σ=0.0042) and MaxEnt (μ=0.9870;σ=0.0021) performed the best in the evaluation phase, followed by GBM (μ=0.9564;σ=0.0159), SVM (μ=0.9516;σ=0.0218), and ANN (μ=0.9349;σ=0.0186).

Figure 2 Performance statistics for evaluation and prediction of models.

Each bar represents the the mean of all the folds from cross validation. (A) Performance statistics related to models built from native range occurrences. (B) Performance statistics related to models built from the California occurrences.

We determined all the models performed well and we produced prediction rasters for each model. Because we were interested in determining our models’ abilities to accurately classify suitable locations, we calculated the percent of correctly defined raster cells when comparing to the estimated naturalized range plot as a measure of model performance. ANN (μ=0.8924;σ=0.0018), MaxEnt (μ=0.8882;σ=0.0037), and GBM (μ=0.8851;σ=0.0089), produced the most accurate predictions followed by RF (μ=0.8823;σ=0.0094), and SVM (μ=0.8751;σ=0.0072).

Different models favored specificity or sensitivity. RF (μ=0.3440;σ=0.1294) performed best with respect to sensitivity, followed by GBM (μ=0.3335;σ=0.1314), SVM (μ=0.2524;σ=0.0590), MaxEnt (μ=0.1484;σ=0.0550), and ANN (μ=0.1319;σ=0.0737). For specificity, our models scored exceptionally well and ranked in the following order: ANN (μ=0.9846;σ=0.0108), MaxEnt (μ=0.9779;σ=0.0108), GBM (μ=0.9520;σ=0.0251), SVM (μ=0.9506;σ=0.0152), and RF (μ=0.9475;σ=0.0580).

Each algorithm was used to create five suitability plots, each pertaining to the model trained on each step in the cross-validation process. While our statistics are associated with the classification/binary plots, we also evaluated the maps of suitability scores (Figs. S1–S5). With respect to our suitability scores, all models favored the western half of the US (roughly 93° W to 124° W) with higher scoring regions around the known naturalized area. All models were consistent across folds. RF, SVM and three GBM models predicted suitability throughout the Appalachian Mountains. RF favored portions of southeastern states Arkansas and Louisiana. Even so, higher suitability scores were prevalent in the western half of the US.

A full analysis of the ensemble performances can be found in Table 1. Statistically, our ensemble models for ANN, GBM, and RF performed similarly to their individual folds and were slightly more accurate and specific (Figs. S6–S10). The performance of the mean and MV ensembles varied by algorithm; however, all five algorithms had improved accuracy via UD ensemble. As for our collective ensembles, collective MV performed well, but ANN MV was more accurate and sensitive. Our collective UD model produced the most convincing result with an accuracy of 0.8973 and the highest sensitivity (0.6918) scores amongst all the ensembles.

Table 1 Ensemble model results for US suitability predictions calibrated from native points vs California points.

Algorithm	Ensemble	Accuracy	Sensitivity	Specificity	
Native points–algorithm ensembles	
ANN	Mean	0.8945	0.1100	0.9896	
ANN	MV	0.8940	0.5488	0.9015	
ANN	UD	0.8936	0.5812	0.8971	
GBM	Mean	0.8898	0.3243	0.9584	
GBM	MV	0.8862	0.4615	0.9202	
GBM	UD	0.8943	0.5350	0.9074	
MaxEnt	Mean	0.8901	0.1373	0.9814	
MaxEnt	MV	0.8884	0.4521	0.9047	
MaxEnt	UD	0.8935	0.5602	0.8979	
RF	Mean	0.8864	0.3224	0.9548	
RF	MV	0.8848	0.4567	0.9230	
RF	UD	0.8932	0.5197	0.9061	
SVM	Mean	0.8613	0.3557	0.9212	
SVM	MV	0.8776	0.3917	0.9119	
SVM	UD	0.8839	0.4123	0.9063	
Collective	MV	0.8917	0.4976	0.9061	
Collective	UD	0.8933	0.6918	0.8941	
California points–algorithm ensembles	
ANN	Mean	0.8968	0.2343	0.9771	
ANN	MV	0.8968	0.5510	0.9144	
ANN	UD	0.8935	0.6080	0.8956	
GBM	Mean	0.8243	0.4227	0.8730	
GBM	MV	0.8043	0.2664	0.9283	
GBM	UD	0.8904	0.4785	0.949	
MaxEnt	Mean	0.8989	0.2113	0.9823	
MaxEnt	MV	0.8984	0.5991	0.9087	
MaxEnt	UD	0.8981	0.6271	0.9048	
RF	Mean	0.8956	0.1956	0.9804	
RF	MV	0.8953	0.5456	0.9084	
RF	UD	0.8979	0.7185	0.9005	
SVM	Mean	0.8837	0.4565	0.9279	
SVM	MV	0.8947	0.5280	0.9142	
SVM	UD	0.8970	0.5871	0.9064	
Collective	MV	0.8978	0.5629	0.9143	
Collective	UD	0.8931	0.8436	0.8932	

Modeling California points

Overall, we saw similar predictions when evaluating models on California points (Fig. 2, Figs. S11–S20). All of our model folds scored greater than 0.99 in AUC with the exception of one ANN fold (0.7121) and one SVM fold (0.9095). We created prediction plots for these models and again calculated the classification accuracy, sensitivity, and specificity for each model. We saw similar accuracy scores for ANN (μ=0.8907;σ=0.0065), MaxEnt (μ=0.8981;σ=0.0465), RF(μ=8714;σ=0.0403), and SVM (μ=0.8926;σ=0.0054) models but lower scores for GBM (μ=0.8167;σ=0.0465).

In comparison to the native range models sensitivity scores were higher for ANN (μ=0.2379;σ=0.1701), GBM (μ=0.4498;σ=0.1939), and MaxEnt (μ=0.2368;σ=0.1248), roughly the same for SVM (μ=0.2576;σ=0.0899), and slightly lower for RF (μ=0.3099;σ=0.2511). Specificity scores were higher only for SVM (μ=0.9696;σ=0.0169), similar for Maxent (μ=0.9783;σ=0.0154) and were lower for ANN (μ=0.9626;σ=0.049), GBM (μ=0.8612;σ=0.0752), and RF (μ=0.9394;σ=0.0753).

We found similar improvements in our ensemble models to that of the individual models (Table 1). Suitable habitat was further reduced to states roughly 103° W to 124° W. The majority of suitability fell in the Chukar naturalized range, and states where Chukars are present especially California, Idaho, Nevada, and Oregon (Fig. 3).

Figure 3 Collective ensemble prediction plots.

Discussion

Pearson & Dawson (2003) state that suitability models favor a hierarchal framework, suggesting that measuring the bioclimatic envelope as a preliminary step in the modeling scheme may give insight to a broad domain of suitability. We limited our models to data derived from climate and topography, which do not speak to localized circumstances and are not exclusive to Chukar. Nonetheless, we were able to produce accurate SDMs using both native range data and data from just a small partition of the US naturalized range. The majority of the SDM plots show a strong suitability in the western half of the US, particularly in states where Chukars exist, confirming that site-level factors are indeed significant predictors of Chukar establishment success.

Traditionally, species occurrences are recorded in field studies or pulled from published databases. Even so, these studies may only represent a portion of the suitable habitat range due to abiotic barriers or individuals simply were not reported in suitable areas. (Phillips et al., 2009; Hirzel & Le Lay, 2008; Robinson et al., 2011). With the growing interest and use of citizen science, SDMs studies now resort to databases such as the Global Biodiversity Information Facility (e.g., Beck et al., 2014; Mori et al., 2019), iNaturalist (e.g., Mori et al., 2019), and eBird (e.g., Steen, Elphick & Tingley, 2019). However, some of these records are unverified as individuals could be misidentified, data may be missing or inaccurate (e.g., longitudinal/latitudinal coordinates reversed), or the inability to differentiate between wild and recently released or escaped captive individuals. To avoid these pitfalls, we implemented a data filtering process on eBird data that limited our study to occurrences with proof of observation, and we removed all duplicate records. Because the distribution of occurrences was consistent with historic range maps (e.g., Christensen, 1970), we found these points suitable for our analysis.

We used algorithms that are widely chosen by ecologists (Drake, Randin & Guisan, 2006; Elith, Leathwick & Hastie, 2008; Elith et al., 2011; Hijmans & Elith, 2017; Hijmans et al., 2017; Schatz, Kramer & Drake, 2017; Norberg et al., 2019) and three classification statistics in our analysis to understand a model’s predictive limitations. With respect to all 84 models produced, we calculated accuracy (μ=0.8823), sensitivity (μ=0.3525), specificity (μ=0.9397). We suspect the low sensitivity scores were due to spatial autocorrelation, which predicts a greater range of suitable location. Even then, our models still favored the states where Chukars are established. These statistics show that while our models perform well in classifying novel locations, the high specificities and lower sensitivities suggest our models are better at predicting locations that are not suitable over those that are. In spite of this, we argue that it is just as important to know this information, especially for gamebird introductions (Nagel, 1945, Bohl, 1957).

Further inspection into species and site-level factors known to affect Chukar introduction success should be considered. Several field studies (Nagel, 1945; Alcorn & Richardson, 1951; Barnett, 1952; Galbreath & Moreland, 1953; Christensen, 1954, 1970, 1996; Bohl, 1957; Harper, Harry & Bailey, 1958) suggest certain land types are associated with successful establishment, though modeling this should be done at a finer spatial scale (Pearson & Dawson, 2003; Thuiller, Araújo & Lavorel, 2004; Luoto, Virkkala & Heikkinen, 2006). It is also well documented that the availability of cheatgrass (Bromus tectorum) is a convincing indicator of establishment in the US (Galbreath & Moreland, 1953; Harper, Harry & Bailey, 1958; Christensen, 1970) though data regarding to cheatgrass occurrence and/or density is minimal, which is why it was excluded from our models. Artificial habitat or watering systems (e.g., ‘guzzlers’) have been integrated into game introductions, which may have been an adequate supplemental resource to facilitate establishment in some studies (Harper, Harry & Bailey, 1958; Christensen, 1970; Larsen et al., 2007). Sub-species environmental tolerance has been shown to impact SDM model predictions (Mori et al., 2019) and while this is implied by our spatial partitioning of the naturalized range and the use of the California data, further studies may be necessary to understand true range potential. Even then, domestication of reared Chukars may also affect introduction attempts (Alcorn & Richardson, 1951).

With this in mind, it is important to note that while our models do not account for specific site-level, species-level, or event level factors, gamebird introduction efforts would benefit from consideration of habitat variables. Modeling of the bioclimatic envelope allows us to not only determine which states consist of potential habitat, but which ones to avoid.

Supplemental Information

Supplemental Information 1 Suitability plots predicted by artificial neural network (ANN) models trained on native points.

Click here for additional data file.

Supplemental Information 2 Suitability plots predicted by gradient boosting trees (GBM) models trained on native points.

Click here for additional data file.

Supplemental Information 3 Suitability plots predicted by maximum entropy (MaxEnt) models trained on native points.

Click here for additional data file.

Supplemental Information 4 Suitability plots predicted by random forest (RF) models trained on native points.

Click here for additional data file.

Supplemental Information 5 Suitability plots predicted by support vector machine (SVM) models trained on native points.

Click here for additional data file.

Supplemental Information 6 Suitability plots predicted by ensembles derived from artificial neural network (ANN) models trained on native points.

Click here for additional data file.

Supplemental Information 7 Suitability plots predicted by ensembles derived from gradient boosting trees (GBM) models trained on native points.

Click here for additional data file.

Supplemental Information 8 Suitability plots predicted by ensembles derived from maximum entropy (MaxEnt) models trained on native points.

Click here for additional data file.

Supplemental Information 9 Suitability plots predicted by ensembles derived from random forest (RF) models trained on native points.

Click here for additional data file.

Supplemental Information 10 Suitability plots predicted by ensembles derived from support vector machine (SVM) models trained on native points.

Click here for additional data file.

Supplemental Information 11 Suitability plots predicted by artificial neural network (ANN) models trained on California points.

Click here for additional data file.

Supplemental Information 12 Suitability plots predicted by gradient boosting trees (GBM) models trained on California points.

Click here for additional data file.

Supplemental Information 13 Suitability plots predicted by maximum entropy (MaxEnt) models trained on California points.

Click here for additional data file.

Supplemental Information 14 Suitability plots predicted by random forest (RF) models trained on California points.

Click here for additional data file.

Supplemental Information 15 Suitability plots predicted by support vector machine (SVM) models trained on California points.

Click here for additional data file.

Supplemental Information 16 Suitability plots predicted by ensembles derived from artificial neural network (ANN) models trained on California points.

Click here for additional data file.

Supplemental Information 17 Suitability plots predicted by ensembles derived from gradient boosting trees (GBM) models trained on California points.

Click here for additional data file.

Supplemental Information 18 Suitability plots predicted by ensembles derived from maximum entropy (MaxEnt) models trained on California points.

Click here for additional data file.

Supplemental Information 19 Suitability plots predicted by ensembles derived from random forest (RF) models trained on California points.

Click here for additional data file.

Supplemental Information 20 Suitability plots predicted by ensembles derived from support vector machine (SVM) models trained on California points.

Click here for additional data file.

Additional Information and Declarations

Competing Interests

Author Contributions

Data Availability

The authors declare that they have no competing interests.

Austin M. Smith conceived and designed the experiments, performed the experiments, analyzed the data, prepared figures and/or tables, authored or reviewed drafts of the paper, and approved the final draft.

Wendell P. Cropper, Jr. conceived and designed the experiments, analyzed the data, authored or reviewed drafts of the paper, and approved the final draft.

Michael P. Moulton conceived and designed the experiments, analyzed the data, authored or reviewed drafts of the paper, and approved the final draft.

The following information was supplied regarding data availability:

Data, code, and supplemental plots are available at GitHub: https://github.com/amsmith8/A-quantitative-assessment-of-site-level-factors-in-influencing-Chukar-introduction-outcomes.

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
