# Peer review of "A quantitative assessment of site-level factors in influencing Chukar (Alectoris chukar) introduction outcomes"

_PeerJ, doi:10.7717/peerj.11280_

## Round 0.1 · original submission · Major Revisions

Dear Austin, Wendell, and Mike

For the time being, I am acting as the sole reviewer of your manuscript on chukars. I want you to make some modest revisions, return the corrected manuscript, and then I will send it our for external review.

As you know I have been an editor or reviewer on many of the papers in this research programme. It’s an important one and generates a substantial discourse from rival factions. It’s important to make your arguments as strong as they can be from the start.

I continue to find the Moulton et al. 2017 paper in Biodiversity and Conservation paper to be the clearest exposition of the problems involved. It shows that most game bird species failed when introduced. That could be the species, the places into which they were released, or the size of the introduction. (Henceforth, we all call that “propagule size.”) Of the species that succeeded, most introductions failed in the places where introduced. That narrows possible causes down to place and propagule size. Of the places where a species does sometimes succeed, there is a weak effect of propagule size. The chukar is one such species. “Places” are states in this discussion. I find this paper to provide the most compelling argument for that effect, despite your critics' well-publicised efforts to show its ubiquity and overarching importance.

Now, your current manuscript has a simple objective. It is to argue that for chukars it is site-level factors that best predict whether chukars succeed or not. You conclude “Our study shows that quantitative models in evaluating environmental variables and that site-level factors are strong 
indicators of introduction success. “

The obvious problem is that outside of your abstract, you don’t discuss “introduction success.” Indeed, you don’t provide explicit justification for the sentence I quoted in the text, tables, or figures. The gold standard would be for you to show your predicted probability of success, based on the various environmental factors you consider, against whether the introduction succeeded or not. You hope to show that it is these factors that predict success rather than propagule size. Indeed, I’d want to see a head-to-head comparison.

As I have done with other papers in your research programme, I am happy to discuss these issues as a sidebar to the formal PeerJ process, allowing us to move quickly toward a manuscript I can submit for external review.

---

## Round 0.2 · Major Revisions

Please see attached file. You still need to address some of my concerns, before I send this out.

---

## Round 0.3 · Minor Revisions

Dear Austin and Mike:

I have struggled to get more than one reviewer. Many don't even reply to one's emails, which I find unconscionably rude. The one review is favourable and, as you know, I have made substantial comments to you on previous versions of this. So, rather than wait any longer, I've made my decision.

The paper does what it proposes, but not what I want it to do. I can't impose the latter on you. But I do feel that you have missed an important chance. You know where these birds have failed. I don't understand why you don't show that based on your model they would be expected to fail there. Doing so, independently of the propagule size, would be a very significant result. Your last sentence claims "Modeling of the bioclimatic envelope allows us to not only determine which states consist of potential habitat, but which ones to avoid." If you have done that, then it's not clear to me.

What I expected — and still hope you can produce —are simple statements that say

"Chukars succeeded in X percent of the places where we predicted suitable habitat, and only Y percent of the places where we predicted unsuitable habitat. In contrast, they failed in P percent of the places where we predicted unsuitable habitat, and succeeded in only Q percent of such places. "

Such statements would make this effort a most powerful rejoinder to your critics.

What I do need you to fix is the statement

"SDMs produced accurate predictions (μ = 0.8823 ) and suitability favored states where Chukars were successfully introduced and are present"

I really have no idea what this means. The four decimal places are at least two too many. But does it mean.

chukars succeeded in 88 percent of the places where we predicted suitable habitat,

and, if so, please say so.

Reviewer 1 ·

Basic reporting

From an Invasion Biology perspective it is interesting that this paper promotes the use of modelling for release of a globally recognised invasive species. I think the manuscript would benefit from a brief species description and invasive status, particularly in North America where this species is promoted. Additionally including the use of a model to predict potential invasive range could broaden the scope of this MS and its impact. In a global context (of interest), introductions in South Africa have also failed, yet a population persists on Robben Island just off the coast. Regardless I think a brief framing of the context of this species (even just for North America) would improve the MS.

Experimental design

Authors have addressed concerns raised by previous reviewers.

Validity of the findings

Findings are clear and well presented. Some clarity on the ebird map is needed. Is this reflecting the authors refined data set or the whole ebird data base? It seems some countries are not represented where known presence of this species is.

Additional comments

Some text in the methods section can be moved to the discussion and there are a few typing errors which can be corrected (in particular spacing between brackets and commas in the methods section lines 216-222). See also lines 86, 89, and 133 for typing errors.

---

## Round 0.4 · accepted · Accept

Dear Austin and Mike: Thanks for your responses. This is another piece in your long and compelling argument for site- and species-level factors determining invasion success. It's a contribution, if not quite the one I was hoping it would be. You justify why it cannot be. Cheers, S.